# A spatial-attentional mechanism underlies action-related distortions of time judgment

Liyu Cao[1,2]*

[1]Department of Psychology and Behavioural Sciences, Zhejiang University, Hangzhou, China; [2]The State Key Lab of Brain-Machine Intelligence, Zhejiang University, Hangzhou, China

*For correspondence:
liyu.cao@zju.edu.cn

Competing interest: The author declares that no competing interests exist.

**Abstract** Temporal binding has been understood as an illusion in timing judgment. When an action triggers an outcome (e.g. a sound) after a brief delay, the action is reported to occur later than if the outcome does not occur, and the outcome is reported to occur earlier than a similar outcome not caused by an action. We show here that an attention mechanism underlies the seeming illusion of timing judgment. In one method, participants watch a rotating clock hand and report event times by noting the clock hand position when the event occurs. We find that visual spatial attention is critically involved in shaping event time reports made in this way. This occurs because action and outcome events result in shifts of attention around the clock rim, thereby biasing the perceived location of the clock hand. Using a probe detection task to measure attention, we show a difference in the distribution of visual spatial attention between a single-event condition (sound only or action only) and a two-event agency condition (action plus sound). Participants accordingly report the timing of the same event (the sound or the action) differently in the two conditions: spatial attentional shifts masquerading as temporal binding. Furthermore, computational modeling based on the attention measure can reproduce the temporal binding effect. Studies that use time judgment as an implicit marker of voluntary agency should first discount the artefactual changes in event timing reports that actually reflect differences in spatial attention. The study also has important implications for related results in mental chronometry obtained with the clock-like method since Wundt, as attention may well be a critical confounding factor in the interpretation of these studies.

## eLife assessment

This **important** paper examined how attention affects temporal binding. Through a combination of careful experimental designs and computational modeling, this study provides **solid** evidence highlighting the role of attention in shaping temporal binding. Overall, the present findings will be of interest to cognitive scientists studying phenomena related to time perception, temporal binding, and spatial attention.

## Introduction

Our own actions influence perception profoundly (*Rolfs and Schweitzer, 2022*). For example, we can barely tickle ourselves, whereas the same tactile stimulation produced by someone else or by an object can be pretty ticklish (*Blakemore et al., 1998*; *Weiskrantz et al., 1971*). In the domain of time judgment, Haggard and colleagues reported an illusion of temporal attraction between an action and a slightly delayed sensory event, known as temporal binding (or intentional binding). Temporal binding is comprised of action binding (i.e. the action being reported as occurring later) and outcome

**Figure 1.** The attention hypothesis of temporal binding. (**a**) Attention in outcome binding. The distribution of attention around the clock rim, at the time close to the event requiring a timing report, receives modulation from both action and action outcome. When the sound time is reported, attention increases only after the onset of the sound in the sound only condition. In the action sound condition, attention is activated prior to the sound onset due to action. The difference in the attention distribution between the two conditions can lead to the difference in the reported clock hand position at the time of sound onset (i.e. outcome binding). (**b**) Attention in action binding. When the action time is reported, the sound in the action sound condition is an extra cue for attention activation compared to the action only condition. Therefore, there is more attention in the action sound condition than the action only condition at clock hand positions after the sound playtime, leading to a later reported clock hand position in the action sound time (i.e. action binding). Please refer to the text for detailed information. A stands for the actual clock hand position when the voluntary keypress is made, and A′ is the reported A from participants. S stands for the actual clock hand position when the sound is played, and S′ is the reported S from participants.

binding (i.e. the sensory event being reported as occurring earlier), with outcome binding having a much bigger effect size (*Haggard et al., 2002*; *Wolpe et al., 2013*). Over the two past decades, the temporal binding effect has attracted cross-disciplinary attention with regard to its cognitive/neural mechanisms and the potential applications especially with its widespread use as an implicit measure of sense of agency (*Antusch et al., 2021*; *Buehner and Humphreys, 2009*; *Dogge et al., 2012*; *Haggard, 2017*; *Kirsch et al., 2019*; *Legaspi and Toyoizumi, 2019*; *Moore and Obhi, 2012*).

However, the impact of visuospatial attention has been heavily overlooked when considering the measurement of time judgment with the widely used clock method (known as the Libet clock method; *Libet et al., 1983*; *Wundt, 1874*). In the standard testing procedure (e.g. *Haggard et al., 2002*), an event was presented (e.g. a keypress or a sound) while participants watched a clock face with a rapidly rotating clock hand. Participants indicated the event time by reporting where the clock hand was positioned at the event onset. This essentially transforms a timing task to a spatial localization task. It is well-known that attention plays an important role in spatial localization (*Fortenbaugh and Robertson, 2011*; *Tse et al., 2011*; *Visser and Enns, 2001*; *Zhou et al., 2016*). When the location of an object is ambiguous, participants tend to localize the object to the place where attention is directed to *Adam et al., 2008*; *Binda et al., 2009*; *Kirsch, 2015*. This is particularly relevant to the temporal binding effect measured with the clock method, as the fast rotating clock hand makes the location report a challenging task (*Haggard and Cole, 2007*).

Outcome binding is usually obtained by comparing an action sound (AS) condition and a sound only (SO) condition. In both conditions, participants report where the clock hand pointed to at the time of sound play. The reported time is earlier in the AS condition than in the SO condition (*Figure 1a*). Suppose that the clock hand is positioned at 12 o'clock when a sound is played (*Figure 1a*). In the SO condition, the sound is controlled by the computer. Since the onset time of the sound is not known in advance, attention may accumulate towards the clock rim only after the sound is played. Therefore, the amount of attention resource is low at the clock hand location before the sound play but high after. In the AS condition, the sound is triggered by an action, which is executed when the clock hand is at around 10 o'clock position (given a sound delay of 250 ms and a revolution period of the clock hand of 1800 ms as employed in the current study). Previous research has provided ample evidence that action modulates the distribution of attention according to the action goal (*Deubel and Schneider, 1996*; *Rolfs et al., 2011*; *Humphreys et al., 1998*). For example, when reaching towards a target, visual attention was shown to be drawn towards the target before the onset of the reaching action (*Baldauf and Deubel, 2010*; *Deubel et al., 1998*; *Eimer et al., 2006*; *Rolfs et al., 2013*). The keypress in the clock method can be successfully performed rather easily without considering the information of a target. However, the timing report task requires the identification of the clock hand position around

the time of the keypress. Conceivably, the task goal (i.e. reporting the clock hand position) may be associated with an attention shift towards the clock rim similar to the attention shift to a target in a target reaching task. This attention shift should result in a high amount of attention resource to the clock hand location before the sound onset in the AS condition (*Figure 1a*). The predicted attention difference between the AS and SO conditions bears a striking resemblance to the timing report difference between the two conditions. Due to the tight link between attention and spatial localization as noted earlier, earlier attention activation in the AS condition compared to the SO condition can lead to an earlier reported clock hand position (i.e. outcome binding).

The attention hypothesis may also be extended to action binding (*Figure 1b*). Action binding is obtained by comparing the AS condition and an AO condition. The reported keypress time is later in the AS condition than in the AO condition. The only difference between the two conditions is that no sound is played after a keypress in the AO condition. The sound in the AS condition may have the effect as an extra source of attention attraction, resulting in a higher amount of attention resource at the clock hand location after the keypress in the AS condition than in the AO condition, which can lead to a later reported keypress time in the AS condition (i.e. action binding).

In a series of four experiments, we demonstrated distinct patterns of attention modulation in temporal binding induced by action and sensory stimulation using the clock method. Furthermore, computational modeling using the attention measure alone can reproduce the temporal binding effect, providing strong supporting evidence for the attention hypothesis of temporal binding.

## Results

### An attention distribution shift in outcome binding

Participants reported the sound onset time using the clock method (*Figure 2*). During the experiment, they were asked to fixate on the center of the visual presentation, but no eye-tracking was employed to ensure compliance with the instruction. A clear outcome binding effect was confirmed in Experiment 1 ($t(17)$ = 9.78, p<0.001, $dz$ = 2.30, one-tailed paired-samples t-test, $BF_{+0}$ = 1.21e6; *Figure 3a*). The reported time was earlier when the sound was triggered by participants through a keypress (AS condition; $M$ = –79.31 ms, *95%* CI = [–102.93 –55.68] ms) than when the sound was controlled by the computer (SO condition; $M$ = 34.17 ms, *95%* CI = [12.73 55.60] ms). Crucially, the pattern of visuospatial attention distribution around the onset of the sound (i.e. the event being judged), operationalized as the probe detection rate, was drastically different between the two conditions (*Figure 3b*). In the SO condition, the detection rate was low before the sound play (referring to both the clock hand locations and the time before the sound play), but high after. In the AS condition, the detection rate was high at the time of keypress, gradually increased, and peaked just before the sound play. After the sound play, the detection rate underwent a sharp decrease. The attention distribution difference was confirmed with a significant interaction effect in a two-way (condition: AS vs. SO; probe location: –50°, –30°, –10°, 10°, 30°, and 50°) within-participants ANOVA comparing the detection rate ($F(5,85)$ = 12.44, p<0.001, $\eta_p^2$ = 0.42, $BF_{incl}$ = 1.87e8). The ANOVA also revealed significant main effects of probe location ($F(5,85)$ = 11.04, p=0.001, $\eta_p^2$ = 0.39, $BF_{incl}$ = 1.15e4) and condition ($F(1,17)$ = 8.17, p=0.011, $\eta_p^2$ = 0.32, $BF_{incl}$ = 3.63).

The above findings were replicated in Experiment 2 with a new group of 20 participants. Experiment 2 was the same as Experiment 1 except that strict eye movement control was applied. During the testing, the eye fixation of participants never exceeded the central area of the clock face (ensured via an eye-tracking device). Again, a clear outcome binding effect was confirmed ($t(13)$ = 6.15, p<0.001, $dz$ = 1.64, one-tailed paired-samples t-test, $BF_{+0}$ = 1.45e3; *Figure 3c*). The reported time was earlier in the AS condition ($M$ = –46.61 ms, *95%* CI = [–85.29 –7.93] ms) than in the SO condition ($M$ = 47.86 ms, *95%* CI = [22.15 73.57] ms). The pattern of visual detection performance was almost identical to that found in Experiment 1 where there was no eye movement control. In the SO condition, attention was low before the sound play, but high after. In the AS condition, attention was high at the time of keypress, gradually increased to its peak just before the sound play, and started to decrease after the sound play (*Figure 3d*). This demonstrates the robustness of the attention modulation induced by action in the AS condition and by the sound in the SO condition. The two-way (condition: AS vs. SO; probe location: –50°, –30°, –10°, 10°, 30°, and 50°) within-participants ANOVA comparing the detection rate confirmed a significant interaction effect ($F(5,65)$ = 4.18, p=0.013, $\eta_p^2$ = 0.24, $BF_{incl}$ =

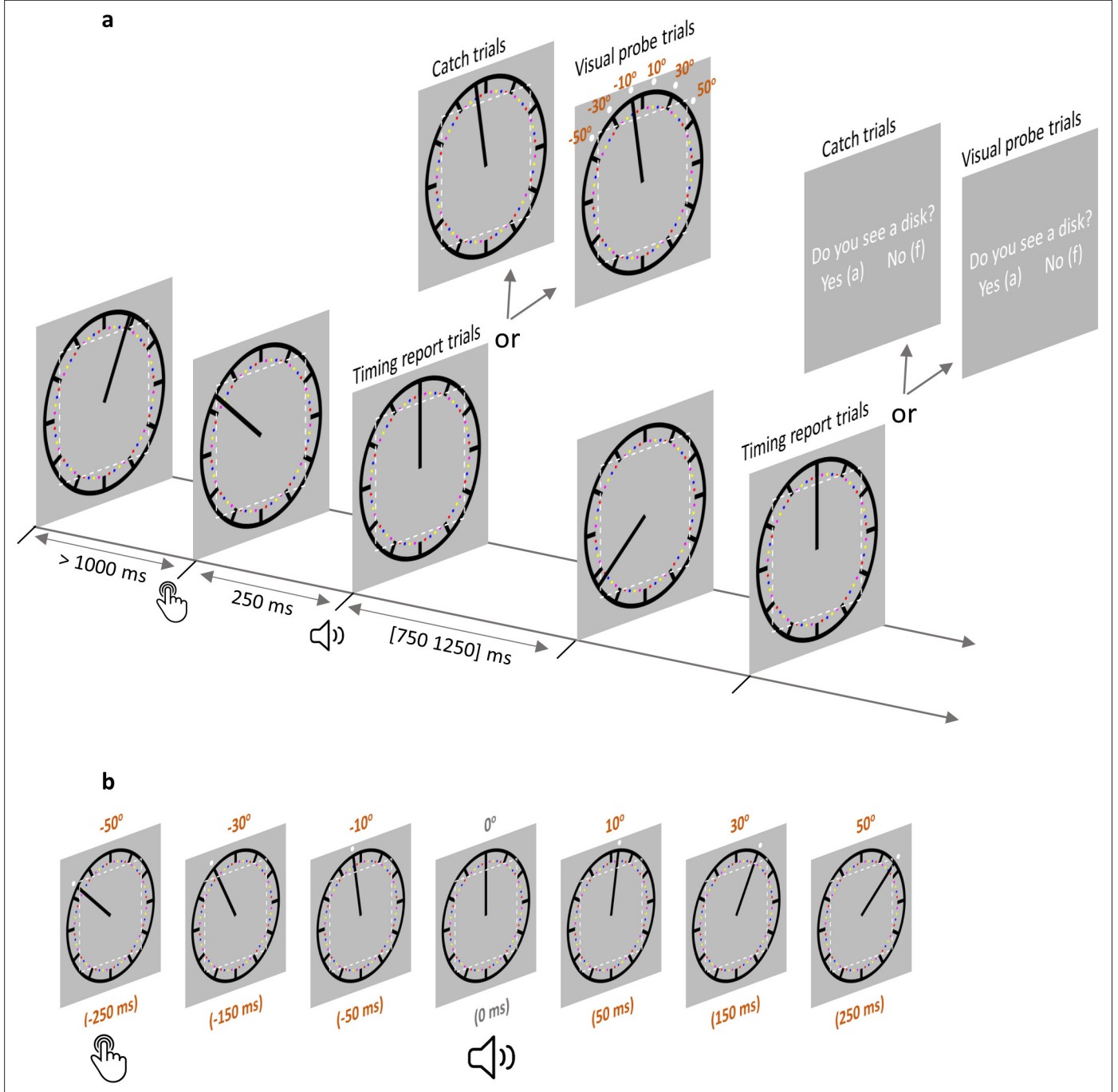

**Figure 2.** Trial structure in the AS condition. (**a**) Each trial started with the clock hand rotating from a random angle. After at least 1000 ms, a voluntary keypress triggered a 250 ms delayed sound. The clock hand continued rotating for another random period between 750 and 1250 ms after sound onset. In timing report trials, participants moved the clock hand back to its position at sound play. In visual probe trials (all six probe locations illustrated) and catch trials, participants reported if a visual probe was detected. The imaginary dotted white square (not shown during the testing) illustrates the eye movement control area in Experiment 2 (eyes moving out of this area would lead to trial abortion). In the SO condition, everything was the same except that no keypress was required. (**b**) A visual probe was presented in each visual probe trial at 1 of 6 possible locations, which also corresponded to six different time points. The position where the clock hand pointed to at the time of sound play was defined as 0° position (0 ms). –50° position (–250 ms) corresponds to the location where the clock hand pointed to when a keypress was made in the AS condition. There were three probe locations before 0° and 3 probe locations after 0°. Note that the visual probe was made salient only for the purpose of illustration. SO: sound only condition; AS: action sound condition.

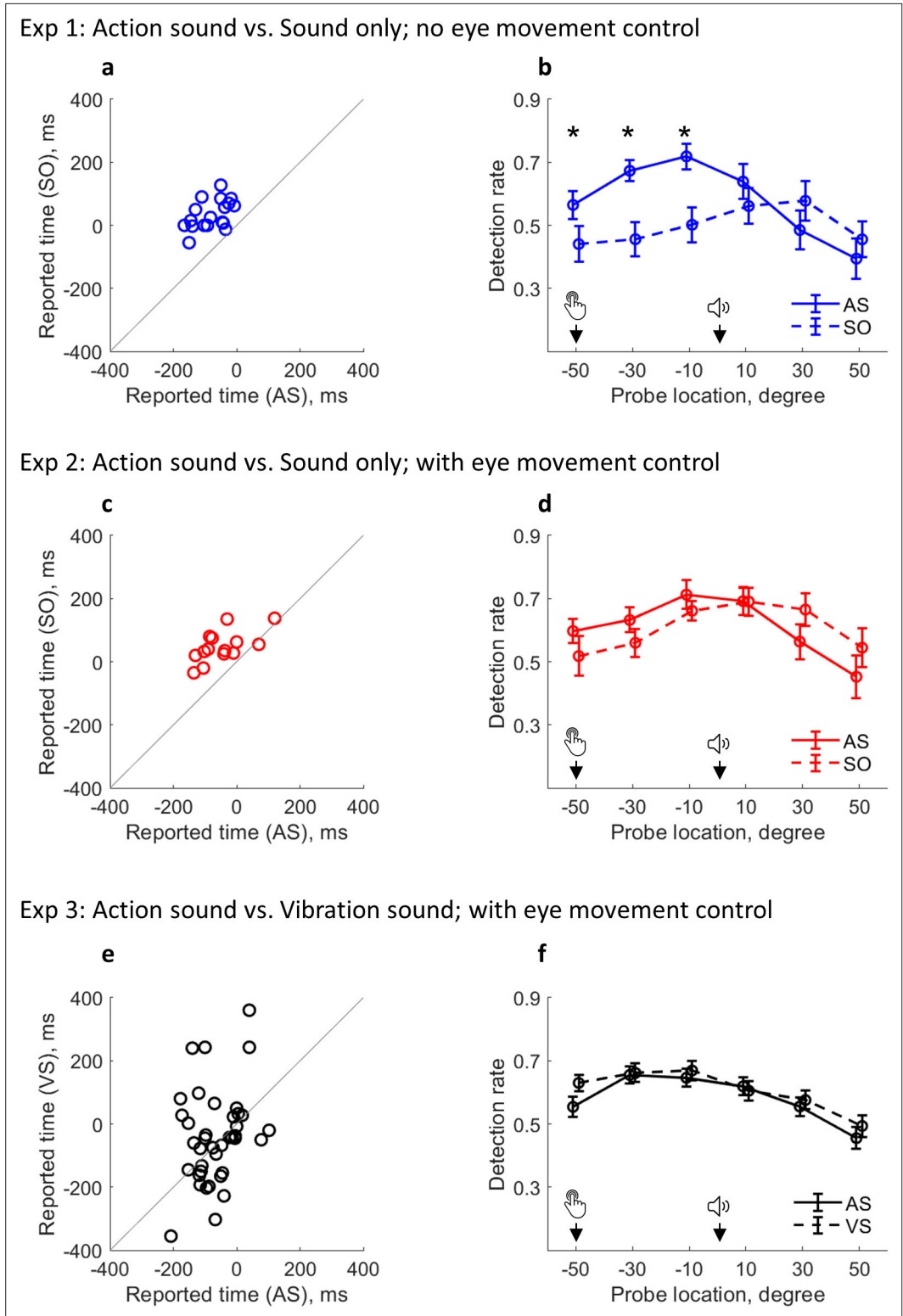

**Figure 3.** Changed attention distribution and its relevance to outcome binding. (**a–b**) Results from Experiment 1. (**a**) Individual sound time report in each condition; (**b**) Detection rate of visual probes as a function of condition and probe location. Attention was activated by the keypress in AS condition and by the sound in the SO condition (note that there was no keypress in the SO condition). Bars represent ±1 standard error (n = 18). Asterisks indicate significant differences between the two conditions at the indicated probe location (false discovery rate adjusted over six comparisons). (**c–d**) Results from Experiment 2 with eye movement control (n = 14). (**e–f**) Results from

*Figure 3 continued on next page*

*Figure 3 continued*

Experiment 3 with eye movement control (n = 39). A VS condition replaced the SO condition. AS: action sound condition; SO: sound only condition; VS: vibration sound condition.

32.91). The ANOVA also revealed a significant main effect of probe location ($F(5,65) = 8.21$, p<0.001, $\eta_p^2 = 0.39$, $BF_{incl} = 2.99e3$) and a non-significant main effect of condition ($F(1,13) = 0.01$, p=0.941, $\eta_p^2 = 0.01$, $BF_{incl} = 0.26$).

## The attention distribution shift is functionally relevant to outcome binding

Experiments 1 and 2 demonstrated the existence of an attention effect in outcome binding. Experiment 3 sought to demonstrate a functional relevance of the attention effect in outcome binding. To this end, the SO condition was replaced with a VS condition, in which a vibrotactile stimulation was applied to the finger 250 ms before the sound play (no keypress). The vibrotactile stimulation had the same timing as the keypress in the AS condition. It should have a similar effect on attention to the keypress in the AS condition, as it is a signal predictive of the sound onset and the timing report task. If this is the case, the binding pattern should vanish when comparing VS and AS conditions.

Another 40 participants were recruited for Experiment 3. Experiment 3 was similar to Experiment 2, but a VS condition replaced the SO condition. Since the crucial evidence here is a null effect (i.e. no statistical difference between VS and AS conditions, or at least the difference between VS and AS conditions being smaller than the difference between SO and AS conditions in Experiments 1 & 2), the sample size was doubled compared to Experiments 1 & 2. Indeed, no binding was found between VS and AS conditions ($t(38) = 1.04$, p=0.305, $dz = 0.17$, two-tailed paired-samples t-test, $BF_{10} = 0.29$; AS condition: $M = -65.19$ ms, 95% CI = $[-87.80 \, -42.59]$ ms; VS condition: $M = -40.96$ ms, 95% CI = $[-87.41 \, 5.49]$ ms; *Figure 3e*). The size of binding in Experiment 3 (as calculated in Experiments 1 & 2) was

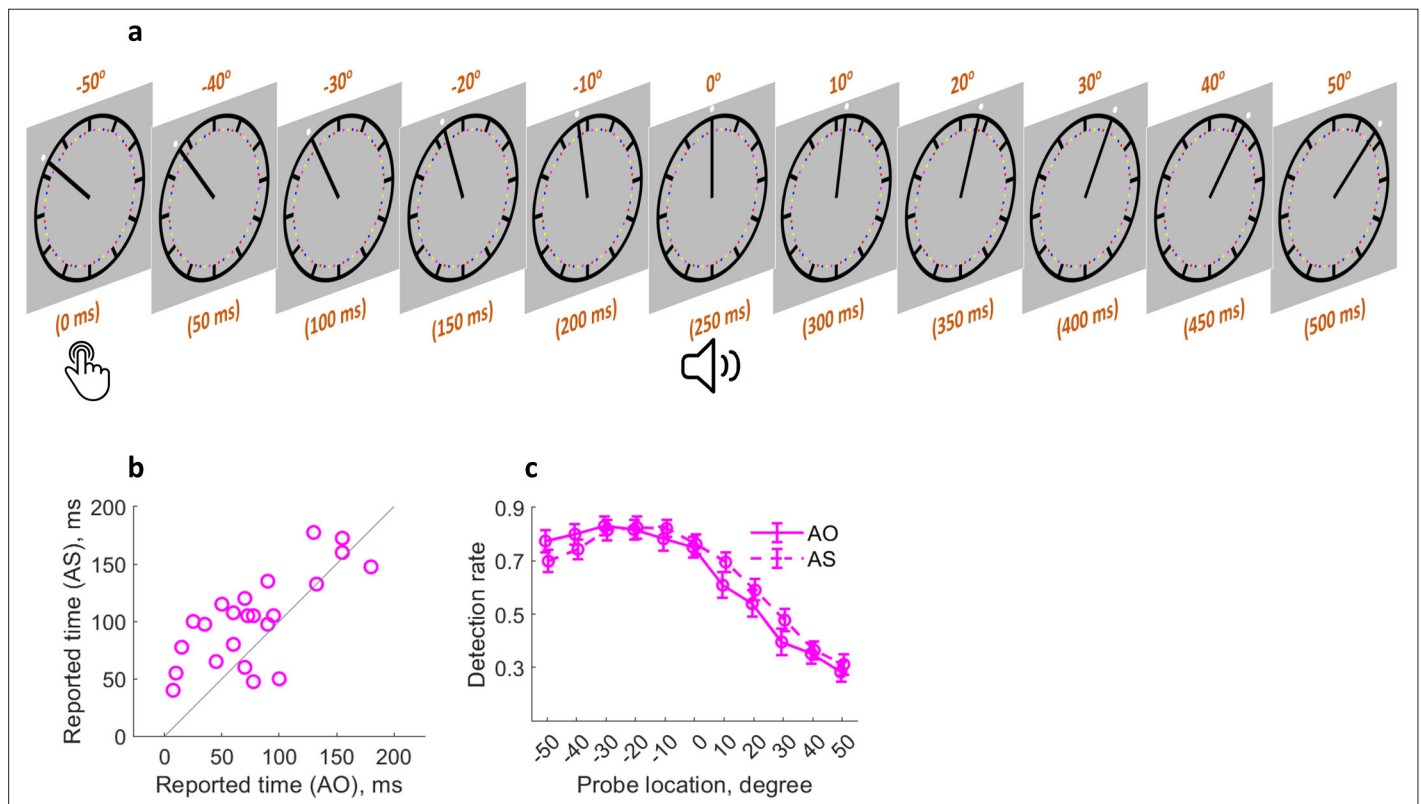

**Figure 4.** Similar attention mechanism in action binding. (**a**) Attention was measured at 11 locations (time points) following the keypress. (**b**) Individual reported keypress time in each condition. (**c**) Detection rate of visual probes as a function of condition and probe location. Bars represent ± 1 standard error (n = 23). AS: action sound condition; AO: action only condition.

smaller than in Experiments 1 & 2 ($t$(69) = –2.99, p=0.002, $ds$ = –0.71, one-tailed unpaired t-test, $BF_{-0}$ = 19.56). The pattern of attention distribution also appeared similar between VS and AS conditions. In both conditions, attention was high before the sound onset, but low after (*Figure 3f*). The two-way (condition: AS vs. VS; probe location: –50°, –30°, –10°, 10°, 30°, and 50°) within-participants ANOVA comparing the detection rate only revealed a significant main effect of probe location ($F$(5,190) = 28.31, p<0.001, $\eta_p^2$ = 0.43, $BF_{incl}$ = 7.17e17). The interaction effect ($F$(5,190) = 1.74, p=0.137, $\eta_p^2$ = 0.04, $BF_{incl}$ = 0.23) and the main effect of condition ($F$(1,38) = 1.34, p=0.254, $\eta_p^2$ = 0.03, $BF_{incl}$ = 0.41) were not significant.

## A similar attention mechanism in action binding

Attention might influence action binding because the action and the sound in the AS (action sound) condition can both be salient cues for attention orientation, especially in the case of action time judgment. Whereas in the AO (action only) condition, no sound was played after the action. Therefore, it is conceivable that the attention resource right after the sound onset should be higher in the AS condition than in the AO condition (*Figure 1b*). Experiment 4 tested the idea of attention involvement in action binding by measuring action binding and attention distribution in the same experiment as similarly done with outcome binding (*Figure 4a*). The action binding effect was confirmed ($t$(22) = 3.45, p=0.001, $dz$ = 0.72, one-tailed paired-samples t-test, $BF_{+0}$ = 34.64; *Figure 4b*). The reported keypress time was later in the AS condition ($M$ = 102.28 ms, 95% CI = [86.06 118.50] ms) than in the SO condition ($M$ = 78.37 ms, 95% CI = [58.91 97.83] ms). The general pattern of attention distribution looks different from that found in the outcome binding task. In both AS and AO conditions, attention was high right after the keypress and then underwent a sharp drop (*Figure 4c*). This is probably due to the fact that the task here was to report the keypress time rather than the sound time. Therefore, attention dropped quickly after the keypress execution. As predicted, there was a significant difference in the attention distribution between the two conditions, which was confirmed by a significant interaction effect in the two-way (condition: AS vs. AO; probe location: –50°, –40°, –30°, –20°, –10°, 0°, 10°, 20°, 30°, 40°, and 50°) within-participants ANOVA comparing the detection rate ($F$(10,220) = 2.93, p=0.026, $\eta_p^2$ = 0.12, $BF_{incl}$ = 18.06). After the sound onset, the amount of attention resource was numerically higher in the AS condition than in the AO condition. The main effect of probe location was unsurprisingly significant ($F$(10,220)=79.56, p<0.001, $\eta_p^2$ = 0.78, $BF_{incl}$ = 1.13e64), and the main effect of condition was not significant ($F$(1,22) = 1.66, p=0.210, $\eta_p^2$ = 0.07, $BF_{incl}$ = 0.25).

## Computational modeling of temporal binding using the attention measure

If attention is critically involved in temporal binding as measured in the current study, it should be theoretically possible to predict temporal binding using the attention measure alone. The attention hypothesis posits that attention is directly related to the timing report. The spatial location receiving more attention should contribute more towards the result of the spatial localization of the clock hand (i.e. timing report). Each location of attention measure corresponded to a potential value of the timing report (e.g. the location where the clock hand pointed to at the time of sound play corresponds to 0 ms in the sound time report task and 250 ms in the keypress time report task). A model was built to predict the individual timing report in each condition through integrating all the measured locations weighted by the attention measure (*Figure 5a*; see also Equations 1-3 in the methods section).

The modeled timing report significantly correlated with the actual timing report in each single condition. In the AS condition of outcome binding (Experiments 1–3), a significant positive correlation was found between the modeled timing report of the sound and the actual timing report of the sound ($r$(65) = 0.55, p<0.001, one-tailed Spearman correlation, 4 outliers removed, $BF_{+0}$ = 4.32e3, *Figure 5b*; without removing outliers: $r$(69) = 0.51, p<0.001, one-tailed Spearman correlation, $BF_{+0}$ = 2.05e3). The same correlation was also found in the SO condition (here the SO condition from Experiments 1 & 2 and the VS condition from Experiment 3 were combined) ($r$(65) = 0.36, p=0.001, one-tailed Spearman correlation, four outliers removed, $BF_{+0}$ = 19.16, *Figure 5c*; without removing outliers: $r$(69) = 0.26, p=0.014, one-tailed Spearman correlation, $BF_{+0}$ = 2.83). For the timing report of the keypress (Experiment 4), a significant correlation was found between the modeled timing report and the actual timing report in both AS condition ($r$(21) = 0.43, p=0.021, one-tailed Spearman correlation, $BF_{+0}$ = 4.78, *Figure 5e*; no outlier detected) and AO condition ($r$(20) = 0.72, p<0.001, one-tailed Spearman

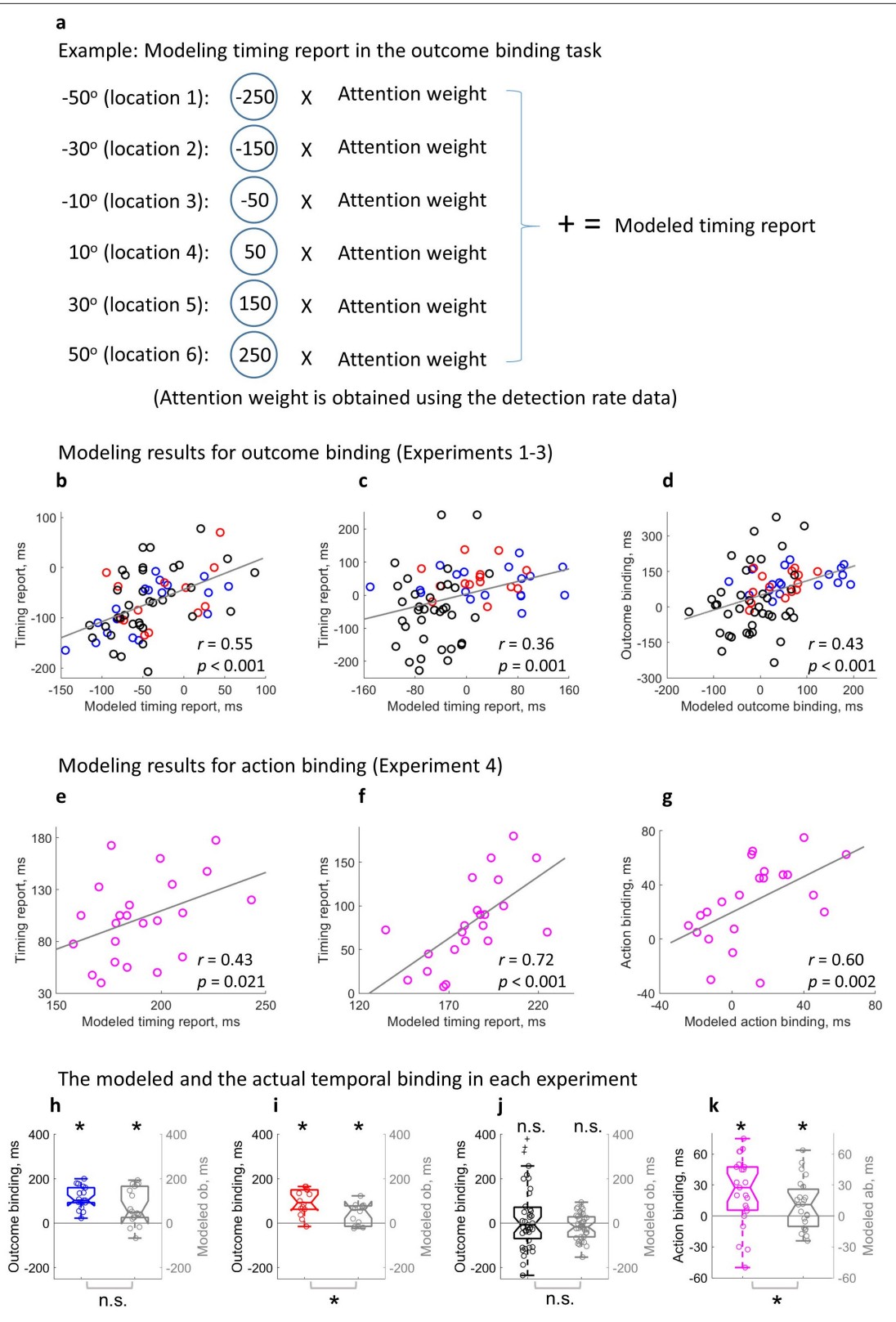

**Figure 5.** Modeling temporal binding with the attention measure. (**a**) Illustration of the timing report modeling in the outcome binding task. (**b–d**) Modeling results for outcome binding (Experiments 1–3 combined). (**b**) Scatter plot of the correlation between the modeled timing report and the actual timing report in the action sound (AS) condition; (**c**) Scatter plot of the correlation between the modeled timing report and the actual timing report in the sound only (SO) and vibration sound (VS) conditions; (**d**) Scatter plot of the correlation between the modeled outcome binding effect and

*Figure 5 continued on next page*

*Figure 5 continued*

the actual outcome binding effect. (**e–g**) Modeling results for action binding (Experiment 4). (**e**) Scatter plot of the correlation between the modeled timing report and the actual timing report in the AS condition; (**f**) Scatter plot of the correlation between the modeled timing report and the actual timing report in the action only (AO) condition; (**g**) Scatter plot of the correlation between the modeled action binding effect and the actual action binding effect. (**h–k**) Box plots of the actual binding effect and the modeled binding effect for all four experiments. Asterisks above indicate significant binding effects. Asterisks below indicate significant differences in the size of actual and modeled binding effects. n.s.: not statistically significant.

correlation, one outlier removed, $BF_{+0}$ = 240.40, *Figure 5f*; without removing outliers: $r(21)$ = 0.74, p<0.001, one-tailed Spearman correlation, $BF_{+0}$ = 391.20).

The modeled temporal binding effect based on attention was obtained by taking the difference of the modeled timing reports in single conditions. For outcome binding (Experiments 1–3), a significant correlation was found between the modeled effect and the actual effect ($r(69)$ = 0.43, p<0.001, one-tailed Spearman correlation, no outliers found, $BF_{+0}$ = 199.22, *Figure 5d*). For action binding (Experiment 4), a similar correlation was found ($r(20)$ = 0.60, p=0.002, one-tailed Spearman correlation, one outlier removed, $BF_{+0}$ = 16.28, *Figure 5g*; without removing outliers: $r(21)$ = 0.60, p=0.001, one-tailed Spearman correlation, $BF_{+0}$ = 17.87).

Lastly, the size of the modeled temporal binding effect was checked for each experiment. In Experiment 1, the modeled outcome binding effect ($M$ = 75.97 ms) was statistically significant ($t(17)$ = 3.98, p<0.001, $dz$ = 0.94, one-tailed paired-samples t-test, $BF_{+0}$ = 77.53; *Figure 5h*), but showed a marginally significant reduction compared to the actual outcome binding effect ($M$ = 113.47 ms; $t(17)$ = –2.01, p=0.061, $dz$ = 0.47, two-tailed paired-samples t-test, $BF_{10}$ = 1.24). The same is true for Experiment 2. In Experiment 2, the modeled outcome binding effect ($M$ = 41.23 ms) was statistically significant ($t(13)$ = 3.22, p=0.003, $dz$ = 0.86, one-tailed paired-samples t-test, $BF_{+0}$ = 15.74; *Figure 5i*), but significantly smaller than the actual outcome binding effect ($M$ = 94.46 ms; $t(13)$ = –3.61, p=0.003, $dz$ = –0.96, two-tailed paired-samples t-test, $BF_{10}$ = 14.61). In Experiment 3, the actual outcome binding effect was not statistically significant due to the attention-matching manipulation. The modeled outcome binding effect was also not significant ($M$ = –14.94 ms, $t(38)$ = 1.61, p=0.116, $dz$ = 0.26, two-tailed paired-samples t-test, $BF_{10}$ = 0.56; *Figure 5j*). Furthermore, there was no significant difference between the modeled and actual effects ($t(38)$ = –1.72, p=0.093, $dz$ = –0.28, two-tailed paired-samples t-test, $BF_{10}$ = 0.66). Importantly, like the actual outcome binding effect showed earlier, the modeled outcome binding effect in Experiment 3 was significantly smaller than the modeled outcome binding in Experiments 1 & 2 ($t(69)$ = –5.00, p<0.001, $ds$ = –1.19, one-tailed unpaired t-test, $BF_{-0}$ = 7.44e3). In Experiment 4, the modeled action binding effect ($M$ = 10.83 ms) was statistically significant ($t(22)$ = 2.16, p=0.021, $dz$ = 0.45, one-tailed paired-samples t-test, $BF_{+0}$ = 2.97; *Figure 5k*), and significantly smaller than the actual action binding effect ($M$ = 23.91 ms; $t(22)$ = –2.17, p=0.041, $dz$ = –0.45, two-tailed paired-samples t-test, $BF_{10}$ = 1.53).

## Discussion

The current study investigated the distribution of visuospatial attention in the temporal binding effect measured with the classic clock method. In four complementary experiments, it was demonstrated that visuospatial attention had a direct link to the event timing report, with the reported time corresponding to the location on the clock where attention was paid to. Action and sensory input both can activate the concentration of attention to the clock, resulting in a dynamic attention distribution pattern around the event which requires a timing report. Critically, the difference in the attention distribution pattern activated by action and sensory input contributes massively to the temporal binding effect. Temporal binding consists of two effects in opposite directions, i.e., a shift of the reported outcome time towards the past in outcome binding, and a shift of the reported action time towards the future in action binding. The attention account provides at least a partial explanation for both outcome binding and action binding in terms of a bias in the distribution of visuospatial attention.

It should be noted that the role of attention in the event timing measure with the clock method has long been speculated (*Haggard and Cole, 2007*; *Libet, 1985*; *Libet et al., 1983*). However, attention in previous studies was discussed more like a kind of general cognitive resources. For example, *Haggard and Cole, 2007* showed that when the event that required a timing judgment at the end of a trial was held in attention (participants knew which event to be judged before the event actually

occurred), the timing report was more accurate than when the event was not held in attention (participants did not know which event to be judged until the event occurred). *Hon, 2023* recently proposed the idea that the attentional resources required to predict action outcomes might be the cause of action binding. In a similar vein, *Schwarz and Weller, 2023* discussed the possibility that the sharing of attentional resources between two events (action and sound) in the instrumental condition resulted in less attention being paid to the event that required timing judgment as compared to the one event baseline condition, thereby leading to temporal binding. Experimental evidence was provided showing that a color change of the clock hand, which was supposed to consume attentional resources, could also lead to the pattern of temporal binding. The present study explicitly pointed out that the visuospatial attention to the clock face had a direct influence on the localization of the clock hand (i.e. the reported event timing). Visuospatial attention was empirically quantified, and its relationship to the timing report in single conditions and the difference of timing reports between two conditions was unequivocally demonstrated.

An immediate question is what temporal binding really means. The canonical view of temporal binding is that it reflects an illusion in timing judgment (*Haggard et al., 2002*; *Tanaka et al., 2019*; *Wen and Imamizu, 2022*). That is, an action appears to occur later when it is followed by a sensory outcome than when not (action binding), and a sensory input seems to occur earlier when it follows an action than when not (outcome binding). In contrast, the data from the current study illustrate the fact that when the event timing judgment is measured with the clock method, there are differences in the contrasting conditions with regards to how the timing judgment is read out from the measurement tool (i.e. the clock). Therefore, the bias in the way of reading measurements poses a challenge to a clear understanding of the variable being measured (i.e. timing judgment). It is unlikely that a changed timing judgment leads to a changed attention distribution, at least in outcome binding. In the outcome binding test, the attention activation following the keypress in the AS condition was before the sound onset, i.e., before the event that requires timing judgment (*Figure 3b and d*). Since timing judgment is a constructive process which integrates available information from both before and after the event of interest (*Cao et al., 2020*; *Haggard et al., 2002*; *Klaffehn et al., 2021*; *Moore and Haggard, 2008*; *Takahata et al., 2012*), there is a theoretical possibility that a common process before the action triggers both a change in attention and a change in timing judgment. One candidate for this common process may be the prediction of the action outcome. However, this idea needs further evidence.

Directly built on the attention hypothesis of temporal binding, computational modeling can reproduce the temporal binding effect using the attention measure alone. This further strengthens the core claim of the current study, i.e., temporal binding is at least severely confounded by visuospatial attention when measured with the clock method. However, the modeled temporal binding effect is smaller than the measured effect (*Figure 5i and k*). This seems to indicate that there is a genuine timing judgment component in temporal binding on top of attention. However, the impact of attention may not have been fully captured in the current study. First, attention was only measured in a limited spatial area (100° starting with the clock hand position at the keypress time). In fact, attention outside this area may also contribute to the timing report. The spatial attention to the locations where the clock hand pointed to at the action preparation stage especially may have a strong influence on the timing report. Due to the unpredictability of the exact action time, measuring the attention before action execution is an experimental challenge. Second, attention may be measured at multiple time points for a given spatial location. We only measured one time point for each spatial location, i.e., the time point when the clock hand was pointing to the specified location. Attention at this time point may have higher relevance to the task compared to other time points, though. This may explain why the size of modeled temporal binding does not match the size of measured temporal binding, i.e., there may be missing attention points that contribute to the timing report. Third, the operationalization of visuospatial attention as probe detection rates does not allow for an accurate quantification of attention differences between different spatial locations. We know that a location with a detection rate of 0.5 has higher level of attention than a location with a detection rate of 0.2, but we do not know by how much (0.5–0.2, 0.5/0.2, or something else?). Without knowing the relationship between attention level and detection rate, the results of the modeled timing report through integrating over spatial locations weighted by the corresponding detection rate may not exactly match the actual timing report, but should at least be proportionate to the actual timing report. Importantly, the strong

and reliable predictive power of attention on the actual timing report in all single conditions and on temporal binding can be demonstrated (*Figure 5b–g*). Experiment 3 provided causal evidence that attention drives the outcome binding effect, as outcome binding was not statistically significant when the attention difference was experimentally controlled for between contrasting conditions (with a doubled sample size). Previous research also showed that temporal binding declined when the temporal predictability of action and effect was controlled for (*Kirsch et al., 2019*). Conceivably, such control allows the attention deployment to operate in a similar manner across different conditions. Therefore, the cognitive underpinnings of temporal binding is currently still an open question.

The present study also has important implications for the clock method in mental chronometry. Wundt was probably the first to use the clock with a fast rotating hand for the scientific study of mental chronometry (*Wundt, 1874*). The method is now also known as the Libet clock method, as Libet popularised this method in his famous study on free will (*Libet et al., 1983*). It is known that the timing report results from this method are quite variable among individuals and strongly depend on the method details such as the speed of the clock hand (*Ivanof et al., 2022*; *Miller et al., 2010*; *Pockett and Miller, 2007*; *Sanford, 1974*; *Seifried et al., 2010*; *Wundt, 1874*; *Yabe and Goodale, 2015*). However, the results are quite often used to draw conclusions about the temporal properties of mental processing. In fact, all these results have involved attention to spatial locations on a clock-like device in order to produce a timing report. The current study suggests that any difference between conditions in the timing report from this method may be in fact due to (or at least confounded by) differences in spatial attention. We should, therefore, investigate when and how such differences in spatial attention might arise in detail in order to better understand the results produced by this method.

Some limitations of the current study need to be born in mind. First, as mentioned before, a full picture of the spatiotemporal visual attention distribution is still waiting to be revealed in the timing report task with the clock method. In the current study, attention was only measured at limited time points and locations. A full picture of the attention distribution pattern could help achieve a complete understanding of the impact of attention on the timing report in the clock method. In addition, a within-participants design paired with an attention intervention paradigm could help determine the impact of attention on temporal binding at the individual level. Second, the clock method may not be an optimal tool for resolving the issue of a timing judgment component in temporal binding. Even if attention could fully account for the temporal binding effect measured with the clock, it could also mean that the timing judgment component could not be easily measured in this way (*Stetson et al., 2006*). The temporal attraction between an action and the action outcome may exist like a Gestalt. However, the Gestalt is lost when measuring its constituent elements (i.e. reporting the time of the action and the outcome separately). Indeed, temporal binding has also been demonstrated using paradigms in which visual attention does not seem to be critically involved, including but not limited to the interval estimation paradigm (*Engbert et al., 2007*; *Humphreys and Buehner, 2009*) and the auditory timer paradigm (*Martinez et al., 2018*; *Muth et al., 2021*). Of course, whether attention in other modalities or other cognitive processes could explain the temporal binding effect reported there is an intriguing question for further investigation.

To sum up, the current study provided novel and important insights into the understanding of temporal binding and time judgment in general measured with the clock method. Attention is critically involved in temporal binding, at least when the measurement process involves attention. It is important to discount the contribution of attention in a clock-like method study before drawing conclusions about the temporal properties in cognition.

## Materials and methods

### Experiment 1

#### Participants

20 participants (11 females; mean age = 21.2, SD = 1.8) were recruited from a local participant pool. Assuming an effect size of 0.89 (note that this is the effect size of outcome binding and action binding combined as reported in a meta-analysis, see *Tanaka et al., 2019*. Since outcome binding has a larger effect size than action binding, the real effect size of outcome binding should be larger than this), 18 participants (two excluded in the formal data analysis, see below) should lead to a statistical power of

0.98 (alpha = 0.05; one-tailed) (**Faul et al., 2007**). All participants have normal or corrected-to-normal vision. Written informed consent was obtained prior to experiment, and participants were debriefed and received monetary payment after the experiment. The experiment was conducted in accordance with the Declaration of Helsinki (2013) and was approved by the Ethics Committee of the Department of Psychology and Behavioural Sciences, Zhejiang University (ethics application number: [2022]003).

## Stimuli, task, and procedure

The experiment consisted of three parts: threshold testing, AS condition, and SO condition, in this order.

In the threshold testing session, the luminance threshold of the visual probe used in AS and SO conditions was obtained using a 2-down-1-up staircase procedure (**Levitt, 1971**). Experimental stimuli were presented on a gray background (RGB value: [128 128 128]; used throughout the experiment). The staircase procedure for obtaining the luminance threshold was run in two parallel lines, with one line having the luminance intensity starting at 128 (RGB value: [128 128 128]; intensity increase line) and the other starting at 200 (RGB value: [200 200 200]; intensity decrease line). Each trial picked a random line until 15 reversals was obtained for each line. In each trial, participants watched a clock face (diameter: 2.7° of visual angle, dva) with a clock hand rapidly rotating clockwise (1800 ms per revolution). The clock hand started rotating from a random angle, and participants were asked to make a keypress ('k' on a standard QWERTY keyboard with the right index finger) no earlier than 1 s from the trial start. A trial would be aborted with a visual warning signal and repeated if it was made before 1 s. After the keypress, a visual probe (a disk with a diameter of 0.1 dva) and a sound (1000 Hz tone, 50 ms long, 5 ms rise/fall envelop, comfortable volume level) were both presented with a delay of 250 ms. The visual probe was presented for 30 ms outside the clock rim (distance to the clock centre: 1.5 dva) but aligned to the position of the clock hand at the onset of the visual probe. After the visual probe presentation, the clock hand continued rotating for a random period between 750 and 1250 ms. Participants were then asked if a visual probe was detected.

In the AS condition, a visual detection task was employed together with the outcome binding measurement using the clock method (**Libet et al., 1983**). Participants were asked to report the time of a sound play (timing report trials) or to report if a visual probe was detected (a visual probe was presented in visual probe trials to assess the distribution of attention; no visual probe was presented in catch trials, for assessing the false alarm rate). In each trial, the clock hand started rotating from a random angle (**Figure 2a**). Participants were asked to make a keypress ('k' on a standard QWERTY keyboard with the right index finger) at their own decision. However, they were told that no strategies should be used to plan the keypress time (e.g. making a keypress when the clock hand was at 3 o'clock position) and that the keypress should not be made within 1 s from the start of the trial. If a keypress was made within 1 s from the start of the trial, a visual warning signal was displayed, and the trial was repeated. After the keypress, a sound was played via the headphones (1000 Hz tone, 50 ms long, 5 ms rise/fall envelop, comfortable volume level) with a 250 ms delay. After the sound, the clock hand continued rotating for a random period between 750 and 1250 ms. In visual probe trials, the threshold titrated visual probe was presented for 30 ms right outside the clock rim (distance to the clock centre: 1.5 dva) close to the top of the clock hand. The location where the clock hand pointed to at the time of sound play was defined as 0° (time 0). Accordingly, the clock hand would point to 10° 50 ms after the sound play, and –10° 50 ms before the sound play, given its rotation speed of 1800 ms per revolution. Visual probes were presented at six possible locations (corresponding to sic time points): –50° (–250 ms; this is when the keypress was made in the AS condition), –30° (–150 ms), –10° (–50 ms), 10° (50 ms), 30° (150 ms), and 50° (250 ms) (**Figure 2b**). In catch trials and timing report trials, no visual probe was presented. At the end of the trial, participants indicated if a visual probe was detected in both visual probe trials and catch trials. In timing report trials, the clock hand always stopped at the 12 o'clock position after the random period succeeding the sound. This is to ensure that the stop position of the clock hand does not introduce any bias in the judgment. Participants moved the clock hand to its position at the time of the sound, using the left hand (pressing 'a' and 's' to move the clock hand counter-clockwise by 10° and 1°, respectively; pressing 'd' and 'f' to move the clock hand clockwise by 1° and 10°, respectively). There were 50 catch trials, 50 timing report trials, and 180 visual probe trials (30 trials for each visual probe location/timing). Trials were presented in a random order. Intermixing all trial types provides the advantage of simultaneous measure of timing report and attention

distribution. The validity of the testing paradigm can be directly assessed through checking the existence of the classic outcome binding effect. Since there were more trials asking for visual probe detection than trials asking for timing report, the participants were told at the beginning of the experiment that they should perform the task as if the timing report were required in each trial, with the aim of ensuring good quality in the timing report. Five practice trials were given before the formal testing.

The SO condition was identical to the AS condition except that the sound play was controlled by the computer (i.e. no keypress was required to trigger the sound). The timing information of the keypress and the stimulus presentation from the AS condition was recorded for its full replication in the SO condition. For this reason, the SO condition always followed the AS condition.

The stimuli were presented on a liquid crystal display screen (refresh rate: 100 Hz; 24-inch screen size). Stimulus generation and presentation was controlled by Psychtoolbox-3 (*Kleiner et al., 2007*) using Matlab (The MathWorks Inc, USA). The sound was presented using a pair of headphones (Beyerdynamic DT 770 pro, 32 OHM, Germany). The experiment was performed in a well-lit, soundproof testing booth.

## Experiment 2

A new group of 20 participants were recruited for Experiment 2 (10 females; mean age = 22.1, SD = 2.7). Assuming an effect size of 0.89, 14 participants (six excluded in the formal data analysis, see below) should lead to a statistical power of 0.93 (alpha = 0.05; one-tailed) (*Faul et al., 2007*). Experiment 2 was the same as Experiment 1 except that a strict eye movement control was employed. In Experiment 1, participants were asked to fixate the centre of the clock face but no measures were taken to enforce this requirement. In Experiment 2, the movements of the right eye were monitored at 1000 Hz with an eye-tracking device (Eyelink Portable Duo, SR Research Ltd, Canada). In all the three parts of testing (threshold testing, AS condition, and SO condition), participants fixated the centre of the clock face. If the right eye was out of a square area of 2.0 dva centering on the clock face at any time from the start of a trial to the point when the clock hand stopped rotating, the trial would be aborted with a visual warning signal and repeated afterwards.

## Experiment 3

A new group of 40 participants were recruited for Experiment 3 (24 females; mean age = 21.9, SD = 2.4). Since the critical evidence in Experiment 3 is a significantly decreased outcome binding effect (to the point even no outcome binding could be found), the sample size was doubled compared to Experiments 1 and 2. This also led to a balanced sample size between Experiment 3 and the combination of Experiments 1 and 2, making an unpaired t-test between the two statistically appropriate. Assuming an effect size of 0.89, 39 participants (1 excluded in the formal data analysis, see below) should lead to a statistical power of >0.99 (alpha = 0.05; one-tailed) (*Faul et al., 2007*). Experiment 3 has the following three changes as compared to Experiment 2. First, the SO condition was replaced by a Vibration Sound condition (VS). In the VS condition, the right index finger (the keypressing finger) received a mild and short vibrotactile stimulation (two impulses in 10 ms) from a miniature electromagnetic solenoid-type stimulator (Dancer Design, UK) 250 ms before the sound onset. The onset of the vibrotactile stimulation aligned with the keypress time in the AS condition. The order of AS and VS conditions was counterbalanced across participants. For participants starting with the VS condition, the onset time of the vibrotactile stimulation was sampled from a normal distribution with the mean and standard deviation taken from the participants starting with the AS condition. Second, the total number of trials in each condition was reduced to 170 (20 catch trials, 30 timing report trials, and 120 visual probe trials with 20 trials for each of the six visual probe locations). Third, the visual fixation area was a circle centered on the clock centre (diameter was the same as the side length of the square in Experiment 2, i.e. 2.0 dva).

## Experiment 4

A new group of 30 participants were recruited for the test of attention in action binding (16 females; mean age = 21.2, SD = 1.9). Two previous studies from our own group using a similar set-up reported an effect size of 0.79 (n=52) (*Cao et al., 2021*) and 0.62 (n=42) (*Cao et al., 2020*). Assuming an effect size of 0.70 here, 23 participants (seven excluded in the formal data analysis, see below) lead to a statistical power of 0.95 (alpha = 0.05; one-tailed) (*Faul et al., 2007*).

Action binding and attention were measured in the same experiment using a similar set-up as in the outcome binding experiments. In the AS condition, participants made a voluntary keypress, which was followed by a 250 ms delayed sound. In timing report trials, the keypress time was reported as the clock hand position at the time of keypress. In visual probe trials, participants reported if a visual probe was detected during the testing. Visual probes were presented at 11 possible locations (corresponding to 11 time points): –50° (0 ms; this is when the keypress was made), –40° (50 ms), –30° (100 ms), –20° (150 ms), –10° (200 ms), 0° (250 ms; this is when the sound was played), 10° (300 ms), 20° (350 ms), 30° (400 ms), 40° (450 ms), and 50° (500 ms) (*Figure 4a*). Here, the keypress time was defined as time 0 as the event of interest in action binding is the keypress. The location definition followed the routine set in the outcome binding experiments. In the AO condition, the keypress was not followed by an auditory outcome. Everything else was the same as the AS condition. There were 480 trials in each condition (100 catch trials, 50 timing report trials, and 330 visual probe trials with 30 trials for each of the 11 visual probe locations). The order of trials was randomized.

The order of the AS and AO conditions was counterbalanced across participants as there is a clear impact of the testing order on the size of action binding (*Cao et al., 2021*). The threshold of the visual probe was obtained prior to the action binding measure using a 2-down-1-up staircase procedure (*Levitt, 1971*). No eye movement control was applied.

## Data analysis
### The same data analysis procedure was applied to the four experiments
The false alarm rate in the catch trials (calculated as the ratio of trials reporting a visual probe was detected) was used for participant exclusion. Four participants were excluded due to extremely high false alarm rate (one from Experiment 1: 0.63; one from Experiment 2: 0.98; two from Experiment 4: 0.69 and 0.89). For the remaining participants, the average false alarm rate was 0.04 ($SD$ = 0.07) in Experiment 1, 0.02 ($SD$ = 0.01) in Experiment 2, 0.06 ($SD$ = 0.08) in Experiment 3, and 0.04 ($SD$ = 0.03) in Experiment 4.

For the timing report trials, the reported time in each trial was calculated as the difference between the reported position of the clock hand and the actual position of the clock hand. This difference in spatial location was converted to a temporal judgment error, based on the clock hand rotation speed of 1800 ms per revolution. The standard deviation and the median of the reported time in each condition (50 trials in each condition for Experiments 1, 2, and 4; 30 trials in each condition for Experiment 3) was used for subject exclusion. Individuals with extreme values of either the standard deviation or the median of reported time were excluded using the median absolute deviation from the median (MAD–median) rule: let p be the individual value and P be the individual values from the whole sample. If |p − median(P)|×0.6745 > 3×MAD–median, this value is an outlier (*Leys et al., 2013*). This procedure further excluded one participant from Experiments 1, five participants from Experiment 2, one participant from Experiment 3, and five participants from Experiment 4. The remaining participants were included in the formal analysis (18 in Experiments 1, 14 in Experiment 2, 39 in Experiment 3, 23 in Experiment 4).

Individual outcome binding effect was calculated as the difference in the median reported sound time between SO and AS conditions (SO − AS; Experiments 1 and 2) or the difference between VS and AS (VS − AS; Experiment 3). Individual action binding effect was calculated as the difference in the median reported keypress time between AS and AO conditions (AS − AO). The group-level outcome binding effect was evaluated with a one-tailed paired-samples t-test comparing the reported sound time between SO and AS conditions for Experiments 1 and 2 (assuming an outcome binding effect), and a two-tailed paired-samples t-test comparing VS and AS conditions for Experiment 3 (assuming the outcome binding effect would be smaller than in Experiments 1 and 2, but not sure to what extent or if the effect could be reversed). The group-level action binding effect was evaluated with a one-tailed paired-samples t-test comparing the reported keypress time between AO and AS conditions (assuming an action binding effect).

The comparison of the outcome binding effect between Experiment 3 and Experiments 1 & 2 was made with one-tailed unpaired t-tests (assuming a smaller effect in Experiment 3). Since no significant difference in outcome binding was found between Experiment 1 and Experiment 2 ($t$(30) = 1.01, p=0.322, $ds$ = 0.36, two-tailed unpaired t-test) and the testing set-up was quite similar, data from Experiments 1 & 2 were combined for the comparison.

For the visual probe trials (with visual probe presentation), the detection rate was calculated for each probe location as the ratio of trials with the probe detected. This was used as a measure of visual attention. The potential attention distribution difference between conditions was evaluated with a two-way (condition and probe location) within-participants ANOVA.

## Computational modeling

For each condition, we performed computational modeling of the timing report results based on the attention distribution pattern of each individual. The rationale is that the reported clock hand position should correspond to the location where attention is directed to. Each of the six locations where attention is measured corresponds to a specific timing report. For example, if the –50° location is reported in the outcome binding experiment, it corresponds to a reported timing of –250 ms. This is because the clock hand is actually at 0° at the sound onset, and the reported timing is the difference between the reported clock hand position and the actual position (50° corresponds to 250 ms given the clock hand speed in the current study). All the six locations were considered in the modeling, with the contribution from each location weighted by its detection rate. To model the timing report in outcome binding (Experiments 1–3) with the attention data, the detection rates were first corrected so that the location with the smallest detection rate did not contribute to the modeled timing report: $D'_k = D_k - \min(D)$ (Equation 1), where $D'_k$ is the corrected detection rate at location k, $D_k$ is the original detection rate at location k, D is the array of all the six original detection rates, and $\min(D)$ is the smallest among the six detection rates. Each location was then weighed using its corrected detection rate (the sum of the weights across the six locations being one) before being multiplied by the location-associated timing report, and then added up: modeled timing report in outcome binding = $\sum_{k=-250}^{250} T_k \times \left( D'_k / \sum_{k=-250}^{250} D'_k \right)$ (Equation 2), where $T_k$ is the reported timing associated with each location (i.e. –250, –150, –50, 50, 150, 250 ms; sound playtime is 0 ms). Similarly, the modeling of timing report in the action binding experiment (Experiment 4) was performed as: modeled timing report in action binding = $\sum_{k=0}^{500} T_k \times \left( D'_k / \sum_{k=0}^{500} D'_k \right)$ (Equation 3), where $T_k$ is the reported timing associated with the 11 attention sampling points (i.e. 0, 50, 100, 150, 200, 250, 300, 350, 400, 450, 500 ms; keypress time is 0 ms). Note that a full cover of the spatiotemporal attention distribution pattern may be necessary for a precise modeling of the reported timing. In the case of action binding, the attention before a keypress was made may be especially important. However, all the attention sampling points were highly relevant to the timing report, as they were close in time and space to the correct clock hand position that should be reported. Another limiting factor in modeling the reported timing is that each location was weighted by its detection rate. Although a high detection rate should be associated with a higher attention than a low detection rate, the exact relationship between detection rate and attention was not known (e.g. does a detection rate difference between 0.4 and 0.2 indicate the same amount of attention difference with a detection rate difference between 0.8 and 0.6?). With all the above well understood, we predicted that the modeled timing report in the current study should at least be proportionate to the empirically measured result. That is, the modeled timing report might not exactly match the actual timing report, but should at least be proportionate to the empirically measured results (e.g. 60% of the actual timing report). A one-tailed Spearman's rank correlation analysis was performed between the modeled timing report and the actual timing report across participants in each single condition, as a positive correlation was predicted. Modeled temporal binding effects were calculated using the modeled timing report in single conditions (SO - AS or VO - AS for modeled outcome binding; AS - AO for modeled action binding). Since positive values were predicted for modeled outcome binding (Experiment 1, 2) and modeled action binding (Experiment 4), one-tailed paired-samples t-test was used. In Experiment 3, the modeled outcome binding may be positive or negative (two-tailed paired-samples t-test), but was predicted to be smaller than in Experiments 1 and 2 (one-tailed unpaired t-test; data in Experiments 1 and 2 were combined). To compare the size of the modeled temporal binding effect and the actual temporal binding effect, a two-tailed paired-samples t-test was used for each experiment. As already noted, the modeled temporal binding effect was predicted to be at least proportionate to the actual temporal binding effect (one-tailed Spearman's rank correlation). In all correlation analyses, bivariate outliers in correlation analyses were detected using the box-plot rule (*Pernet et al., 2012*), and excluded afterwards. However, results without outlier exclusion were also reported for reference.

## Bayesian statistics

Along with the traditional frequentist statistics, Bayesian statistical results obtained with JASP (version 0.17.2.1) were reported (*JASP Team, 2023*). The default prior setting from JASP was used. For the ANOVA, matched models were used to assess the effects. For the correlation analysis, Kendall's rank correlation was used as Spearman's rank correlation was not available in JASP.

## Acknowledgements

We would like to thank Patrick Haggard for the insightful discussions and helpful comments on an earlier draft, Wilfried Kunde for helpful comments on an earlier draft, and Junyi Dai for statistical consultations. This work was supported by the National Natural Science Foundation of China (grant number: 32271078) and the STI 2030—Major Projects (grant number: 2021ZD0200409).

## Additional information

### Funding

| Funder | Grant reference number | Author |
|---|---|---|
| National Natural Science Foundation of China | 32271078 | Liyu Cao |
| Ministry of Science and Technology of the People's Republic of China | STI 2030-Major Projects 2021ZD0200409 | Liyu Cao |

The funders had no role in study design, data collection and interpretation, or the decision to submit the work for publication.

### Author contributions

Liyu Cao, Conceptualization, Resources, Data curation, Software, Formal analysis, Funding acquisition, Validation, Investigation, Visualization, Methodology, Writing - original draft, Project administration, Writing - review and editing

### Author ORCIDs

Liyu Cao http://orcid.org/0000-0002-1124-9579

### Ethics

Written informed consent was obtained from the participants prior to experiment. The study was approved by the Ethics Committee of Department of Psychology and Behavioural Sciences, Zhejiang University (ethics application number: [2022]003). We have conformed with the Helsinki Declaration of 1975 (as revised in 2013) concerning Human and Animal Rights during the experiments.

Reviewer #1 (Public Review): https://doi.org/10.7554/eLife.91825.3.sa1
Reviewer #2 (Public Review): https://doi.org/10.7554/eLife.91825.3.sa2
Author Response https://doi.org/10.7554/eLife.91825.3.sa3

## Additional files

### Supplementary files
• MDAR checklist

### Data availability

The original data and Matlab analysis code are freely available from Figshare (https://doi.org/10.6084/m9.figshare.23917062).

The following dataset was generated:

| Author(s) | Year | Dataset title | Dataset URL | Database and Identifier |
|---|---|---|---|---|
| Cao L | 2023 | Data for 'A spatial-attentional mechanism underlies action-related distortions of time perception' | https://doi.org/10.6084/m9.figshare.23917062.v1 | figshare, 10.6084/m9.figshare.23917062.v1 |

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
