## [Editor Report · eLife assessment]

This **important** paper examined how attention affects temporal binding. Through a combination of careful experimental designs and computational modeling, this study provides **solid** evidence highlighting the role of attention in shaping temporal binding. Overall, the present findings will be of interest to cognitive scientists studying phenomena related to time perception, temporal binding, and spatial attention.

---

## [Referee Report · Reviewer #1 (Public Review)]

Summary: This study addressed an alternative hypothesis to temporal binding phenomena. In temporal binding, two events that are separated in time are "pulled" towards one another, such that they appear more coincidental. Previous research has shown evidence of temporal binding events in the context of actions and multisensory events. In this context, the author revisits the well-known Libet clock paradigm, in which subjects view a moving clock face, press a button at a time of their choosing to stop the clock, a tone is played (after some delay), and then subjects move the clock dial to the point where the one occurred (or when the action occurred). Classically, the reported clock time is a combination of the action and sound times. The author here suggests that attention can explain this by a mechanism in which the clock dial leads to a roving window of spatiotemporal attention (that is, it extends in both space and time around the dial). To test this, the author conducted a number of experiments where subjects performed the Libet clock experiment, but with a variety of different stimulus combinations. Crucially, a visual detection task was introduced by flashing a disc at different positions along the clock face. The results showed that detection performance was also "pulled" towards the action event or sensory event, depending on the condition. A model of roving spatiotemporal attention replicated these effects, providing further evidence of the attentional window.

The study provides a novel explanation for temporal binding phenomena, with clear and cleverly designed experiments. The results provide a nice fit to the proposed model, and the model itself is able to recapitulate the observed effects.

---

## [Referee Report · Reviewer #2 (Public Review)]

Summary:

Temporal binding, generally considered a timing illusion, results from actions triggering outcomes after a brief delay, distorting perceived timing. The present study investigates the relationship between attention and the perception of timing by employing a series of tasks involving auditory and visual stimuli. The results highlight the role of attention in event timing and the functional relevance of attention in outcome binding.

Strengths:

- Experimental Design: The manuscript details a well-structured sequence of experiments investigating the attention effect in outcome binding. Thoughtful variations in manipulation conditions and stimuli contribute to a thorough and meaningful investigation of the phenomenon.

- Statistical Analysis: The manuscript employs a diverse set of statistical tests, demonstrating careful selection and execution. This statistical approach enhances the reliability of the reported findings.

- Narrative Clarity: Both in-text descriptions and figures provide clear insights into the experiments and their results, facilitating readers in following the logic of the study.

Weaknesses:

- Conceptual Clarity: The manuscript aims to integrate key concepts in human cognitive functions, including attention, timing perception, and sensorimotor processes. However, before introducing experiments, there's a need for clearer definitions and explanations of these concepts and their known and unknown interrelationships. Given the complexity of attention, a more detailed discussion, including specific types and properties, would enhance reader comprehension.

- Computational Modeling: The manuscript lacks clarity in explaining the model architecture and setup, and it's unclear if control comparisons were conducted. These details are critical for readers to properly interpret attention-related findings in the modeling section. Providing a clearer overview of these aspects will improve the overall understanding of the computational models used.

---

## [Author Response]

The following is the authors’ response to the original reviews.

**Public Reviews:**

**Reviewer #1 (Public Review):**
Summary:This study addressed an alternative hypothesis to temporal binding phenomena. In temporal binding, two events that are separated in time are "pulled" towards one another, such that they appear more coincidental. Previous research has shown evidence of temporal binding events in the context of actions and multisensory events. In this context, the author revisits the well-known Libet clock paradigm, in which subjects view a moving clock face, press a button at a time of their choosing to stop the clock, a tone is played (after some delay), and then subjects move the clock dial to the point where the one occurred (or when the action occurred). Classically, the reported clock time is a combination of the action and sound times. The author here suggests that attention can explain this by a mechanism in which the clock dial leads to a roving window of spatiotemporal attention (that is, it extends in both space and time around the dial). To test this, the author conducted a number of experiments where subjects performed the Libet clock experiment, but with a variety of different stimulus combinations. Crucially, a visual detection task was introduced by flashing a disc at different positions along the clock face. The results showed that detection performance was also "pulled" towards the action event or sensory event, depending on the condition. A model of roving spatiotemporal attention replicated these effects, providing further evidence of the attentional window.Strengths:The study provides a novel explanation for temporal binding phenomena, with clear and cleverly designed experiments. The results provide a nice fit to the proposed model, and the model itself is able to recapitulate the observed effects.Weaknesses:Despite the above, the paper could be clearer on why these effects are occurring. In particular, the control experiment introduced in Experiment 3 is not well justified. Why should a tactile stimulus not lead to a similar effect? There are possibilities here, but the author could do well to lay them out. Further, from a perspective related to the attentional explanation, other alternatives are not explored. The author cites and considers work suggesting that temporal binding relies on a Bayesian cue combination mechanism, in which the estimate is pulled towards the stimulus with the lowest variance, but this is not discussed. None of this necessarily detracts from the findings, but otherwise makes the case for attention less clear.

I would like to thank the reviewer for the helpful comments and recommendations.Regarding Experiment 3, the rationale is this. We showed in Experiments 1 and 2 that, for outcome binding, there were two types of difference between Action Sound condition and Sound Only condition: the reported time of sound onset (i.e. the reported clock hand location at the sound onset) and the attention distribution. To experimentally test the relevance of the attention difference to the difference of reported time, we created a situation where the attention difference could be minimised and then checked the difference of reported time. We found that when the attention difference was controlled for between the two conditions, the difference of reported time was also gone, thus providing further evidence for a close link between attention and time report in the current testing paradigm. Therefore, Experiment 3 was primarily targeting the experimental evidence for the claim of the current study. What we needed in Experiment 3 was a condition that could have a smaller attention difference with the Action Sound condition than the attention difference between Sound Only and Action Sound conditions in Experiments 1 and 2. We expected that a tactile stimulus before the sound onset could work, without a clear prediction of the strength of the tactile stimulus in shifting attention, which was also not necessary. This experimental manipulation was a nice fit for the purpose of experiment 3, as we could empirically measur the effectiveness of the tactile stimulus on attention shift and then relate it to the changes in outcome binding.

As the reviewer correctly suggested, the Bayesian framework has been applied in several studies to explain the time judgement distortion in sensorimotor situations (e.g. the temporal binding effect studied here). However, the current study asked what temporal binding is really about when it is measured with the Libet clock method. Is it really about a distortion in time perception (which the Bayesian account tries to explain)? Or is it also about attention? The results showed that the spatiotemporal attention distribution is at least a confound in measuring the perceived time of an event using the Libet clock method. Therefore, the Bayesian account raised in previous studies is relevant when explaining the distortion in time perception, given that it really exists. We here asked if the distortion really exists, and to what extent.

**Reviewer #2 (Public Review):**
Summary:Temporal binding, generally considered a timing illusion, results from actions triggering outcomes after a brief delay, distorting perceived timing. The present study investigates the relationship between attention and the perception of timing by employing a series of tasks involving auditory and visual stimuli. The results highlight the role of attention in event timing and the functional relevance of attention in outcome binding.Strengths:Experimental Design: The manuscript details a well-structured sequence of experiments investigating the attention effect in outcome binding. Thoughtful variations in manipulation conditions and stimuli contribute to a thorough and meaningful investigation of the phenomenon.Statistical Analysis: The manuscript employs a diverse set of statistical tests, demonstrating careful selection and execution. This statistical approach enhances the reliability of the reported findings.Narrative Clarity: Both in-text descriptions and figures provide clear insights into the experiments and their results, facilitating readers in following the logic of the study.Weaknesses:Conceptual Clarity: The manuscript aims to integrate key concepts in human cognitive functions, including attention, timing perception, and sensorimotor processes. However, before introducing experiments, there's a need for clearer definitions and explanations of these concepts and their known and unknown interrelationships. Given the complexity of attention, a more detailed discussion, including specific types and properties, would enhance reader comprehension.Computational Modeling: The manuscript lacks clarity in explaining the model architecture and setup, and it's unclear if control comparisons were conducted. These details are critical for readers to properly interpret attention-related findings in the modeling section. Providing a clearer overview of these aspects will improve the overall understanding of the computational models used.

I would like to thank the reviewer for the helpful comments and recommendations.The attention in the current study, which has been made clearer in the revised manuscript, refers specifically to visuospatial attention. It is presented as a key factor shaping the results of timing report obtained with the clock method, thereby contributing to the explanation of temporal binding. Indeed, attention has been mentioned previously in a similar context, but was treated vaguely as a kind of general cognitive resources. The current study specifically tested and verified that the visuospatial attention paid to the clock face influenced the timing reports. This point has been discussed in a dedicated paragraph in the discussion section of the revised manuscript.

The modeling of the timing report using the attention data was based on a very simple idea: The clock hand location receiving more attention should be given more weight when participants made the timing report (i.e. reporting the clock hand position). The weight for each location was calculated using the detection rate at each location. The relevant methods section has been extensively revised to provide a step-by-step implementation of the modeling, with rationales and pitfalls in the interpretation of the modeling results given (also in the discussion section).